# Reversible host cell surface remodelling limits immune recognition and maximizes survival of *Plasmodium falciparum* gametocytes

Priscilla Ngotho[1,2], Kathleen Dantzler Press[3], Megan Peedell[1,2], William Muasya[1],
Brian Roy Omondi[4], Stanley E. Otoboh[4], Jahiro Gomez[5], Lorena Coronado[5], Karl B. Seydel[6,7],
Melissa Kapulu[8], Miriam Laufer[9], Terrie Taylor[6,7], Teun Bousema[10], Matthias Marti[1,2]*

1 Wellcome Centre for Integrative Parasitology, University of Glasgow, Glasgow, United Kingdom,
2 Institute of Parasitology, Vetsuisse Faculty, University of Zurich, Zurich, Switzerland, 3 Department
of Medicine, Stanford University School of Medicine, Stanford, California, United States of America,
4 Institute of Immunology and Infection Research, School of Biological Sciences, University of Edinburgh,
Edinburgh, United Kingdom, 5 Instituto de Investigaciones Científicas y Servicios de alta Tecnología de
Panamá, Panamá City, Panamá, 6 Department of Osteopathic Medical Specialties, College of Osteopathic
Medicine, Michigan State University, East Lansing, Michigan, United States of America, 7 Blantyre Malaria
Project, Kamuzu University of Health Sciences, Blantyre, Malawi, 8 KEMRI Wellcome Trust Research
Programme, Kilifi, Kenya, 9 Center for Vaccine Development and Global Health, University of Maryland
School of Medicine Baltimore, Maryland, United States of America, 10 Radboud Institute for Health
Sciences, Radboud University Medical Center, Nijmegen, Netherlands

* Matthias.Marti@glasgow.ac.uk

UNITED STATES OF AMERICA

**Peer Review History:** PLOS recognizes the
benefits of transparency in the peer review
process; therefore, we enable the publication
of all of the content of peer review and
author responses alongside final, published
articles. The editorial history of this article is
available here: https://doi.org/10.1371/journal.
ppat.1013110

## Abstract

Reducing malaria transmission has been a major pillar of control programmes and is
considered crucial for achieving malaria elimination. Gametocytes, the transmissible
forms of the *P. falciparum* parasite, arise during the blood stage of the parasite and
develop through 5 morphologically distinct stages. Immature gametocytes (stage I-IV)
sequester and develop in the extravascular niche of the bone marrow and possibly
spleen. Only mature stage V gametocytes re-enter peripheral circulation to be taken
up by mosquitoes for successful onward transmission. We have recently shown that
immature, but not mature gametocytes are targets of host immune responses and
identified putative target surface antigens. We hypothesize that these antigens play
a role in gametocyte sequestration and contribute to acquired transmission-reducing
immunity. Here we demonstrate that surface antigen expression, serum reactivity
by human IgG, and opsonic phagocytosis by macrophages all show similar dynam-
ics during gametocyte maturation, i.e., peaking in the immature stages and taper-
ing off in mature gametocytes. Moreover, the switch in surface reactivity coincides
with reversal in phosphatidylserine (PS) surface exposure, a marker for red blood
cell age and clearance. PS is exposed on the surface of a proportion of immature
gametocyte-infected RBCs (as well as in late asexual stages) but is removed from
the surface in later gametocyte stages (IV-V). Using parasite reverse genetics and
drug perturbations, we confirm that parasite protein export into the host cell and

**Data availability statement:** All relevant data are within the paper and its Supporting Information files.

**Funding:** This work was supported by ERC Consolidator award BoneMalar (PN, KDP, MP, MM), ERC Consolidator award 864180 (TB), MRC Programme grant MR/T016272/1 (PN, MP, MM, TB), Royal Society Wolfson Merit Award (MM), Wellcome Trust Center award 104111 (PN, KDP, MP, MM), NIH ICEMR grant U19AI129388 (TT) and Secretaría Nacional de Ciencia, Tecnología e Innovación (SENACYT) Grant DDCCT 204-2023 (JG, LC). The funders had no role in study design, data collection and analysis, decision to publish, or preparation of the manuscript.

**Competing interests:** The authors have declared that no competing interests exist.

phospholipid scramblase activity are required for the observed surface modifications in asexual and sexual *P. falciparum* stages. Based on these findings we propose that the reversible surface remodelling allows (i) immature gametocyte sequestration in bone marrow followed by (ii) mature gametocyte release into peripheral circulation (and immune evasion due to loss of surface antigens), therefore contributing to mature gametocyte survival *in vivo* and onward transmission to mosquitoes. Importantly, blocking scramblase activity during gametocyte maturation results in efficient clearance of mature gametocytes, revealing a potential path for transmission blocking interventions. Our studies have important implications for our understanding of parasite biology and form a starting point for novel intervention strategies to simultaneously reduce parasite burden and transmission.

## Author summary

Malaria remains a major global health problem and elimination of this global killer is a top priority. Parasite transmission from human to mosquito is critical for disease spread and hence a major target of interventions. Transmission stages (or gametocytes) develop within red blood cells in the immune protected extravascular spaces of the bone marrow and undergo a series of morphological and biophysical changes. Immature gametocytes are rigid and expose antigens on the red blood cell (RBC) surface, presumably to keep the parasite out of circulation during its maturation. In contrast, mature gametocytes are immunologically silent while they passage through the peripheral circulation in preparation for uptake during a mosquito bite. Here we show that reversible surface antigen exposure during gametocyte maturation coincides with transient flipping of phosphatidylserine (PS), a major component of the RBC membrane, onto the RBC surface. This process depends on a parasite scramblase, while the later PS removal requires a host scramblase. Both antigen and PS exposure can induce parasite clearance by macrophages. When blocking host scramblase activity with inhibitors, PS remains on the RBC surface of mature gametocytes providing a rationale for novel transmission blocking interventions.

## Introduction

*Plasmodium falciparum* malaria remains a devastating global health burden despite concerted efforts towards control and elimination. Recent health service disruptions due to the COVID-19 pandemic resulted in a reversal of progress in reducing mortality and morbidity made in the last decade [1]. The most effective approaches to curb malaria, vector control and prompt diagnosis and treatment, are challenged by insecticide and drug resistance. Their dependence on efficient health systems in economically and socio-political stable environments further highlights the need for additional approaches towards elimination in low-resource settings.

An essential component of malaria elimination is interrupting the transmission cycle from human to vector and back to human. Therefore, blocking transmission from human to vector is an attractive target for novel therapeutics. In the human host, *P. falciparum* cycles through various host cells and organs and extensively modifies these host cells for its metabolic needs for replication and transmission. During its blood stage cycle, *P falciparum* infects naïve red blood cells (RBCs) and replicates asexually every 48h hours, resulting in cyclic fevers to the host as well as other clinical manifestation of disease. A sub population of parasites differentiate to the sexual replicative forms (gametocytes) that are the only forms capable of transmitting to mosquitoes. Gametocytes primarily develop in hematopoietic niches of bone marrow and possibly spleen, a feature that is conserved across the *Plasmodium* lineage [2–8]. A unique feature of *P. falciparum* gametocytes is the prolonged development cycle that coincides with morphological differentiation delineating the 5 stages (I-V) of gametocytogenesis [9]. During this gametocyte maturation period, the host cell undergoes profound biophysical changes. Immature sequestering forms become rigid [10,11] and express antigens on the infected red blood cell (iRBC) surface [12]. During maturation from stage IV to V, parasites switch into a deformable state without antigens on the iRBC surface. Hence, these deformable mature forms are immunologically silent, allowing release into circulation while avoiding both immune clearance and splenic filtration. Surface antigens play a key role in sequestration of mature asexual parasites and are therefore hypothesized to mediate gametocyte sequestration as well. Immature gametocytes lack key features on the iRBC surface mediating asexual parasite sequestration, in particular knob structures and the major surface antigen and cytoadherence ligand PfEMP1 [13,14]. Therefore, the mode of interaction with other cells in the hematopoietic niche remains unclear. We have previously identified potential surface antigens that may play this role, including GEXP07 and GEXP10 [12,15,16]. Importantly, once parasites are in circulation, surface exposed parasite antigens trigger immune clearance mechanisms such as opsonic phagocytosis [12,17,18]. Therefore, it would be in the parasites' interest to ensure these modifications only mark the iRBC surface while outside of circulation.

We hypothesise that temporary iRBC surface modifications during gametocyte development enable efficient sequestration of immature stages, while its reversal facilitates mature gametocyte exit from the hematopoietic niche and "safe" circulation in peripheral blood without detection and clearance by host immunity. Perturbation of this reversible process could prevent parasite sequestration or subsequent release into circulation, and/or expose immunogenic targets and reduce onward transmission of the parasite to the mosquito vector. Here, we have tested the hypothesis and demonstrate that reversible surface antigen exposure coincides with a reversible flipping of phosphatidylserine onto the iRBC surface, a process that can be blocked to trigger efficient phagocytosis of the otherwise immunologically "silent" mature gametocyte.

## Results

### Dynamics of surface antigen exposure in *P. falciparum* gametocytes

We recently carried out an analysis of surface antigen expression in *P. falciparum* gametocytes using a transgenic line (Pf2004 Tdtom) that expresses a fluorescent reporter starting at approximately 30 hours of gametocytogenesis [10,12]. This study revealed that immune sera as well as specific antibodies raised against gametocyte surface antigens labelled the surface of immature (Stage II/III) but not mature (Stage V) gametocyte-infected red blood cells (giRBC) [12]. To further define the dynamics of antigen expression, we carried out a time course during gametocyte maturation. For this purpose, we used both the Tdtom reporter line and a second reporter line that expresses a GFP-tagged version of the exported GEXP02 antigen [19,20]. This antigen is expressed from early gametocyte development (i.e., gametocyte ring) and stays on throughout the sexual development cycle [19,20]. Using a combination of flow cytometry and fluorescent microscopy, we measured giRBC surface recognition using a pool of highly reactive patient serum. We observed minimal surface recognition in gametocyte rings, followed by a gradual increase peaking at stage II gametocytes (Fig 1A and 1B). In asexual stages, a similar pattern was observed with minimal signal in ring stages and major recognition in schizonts (Figs 1A, 1B, and S1G). After the peak at stage II, giRBCs lost surface antigen expression as they matured with minimal signal observed in stage V. We also measured surface reactivity against two iRBC surface

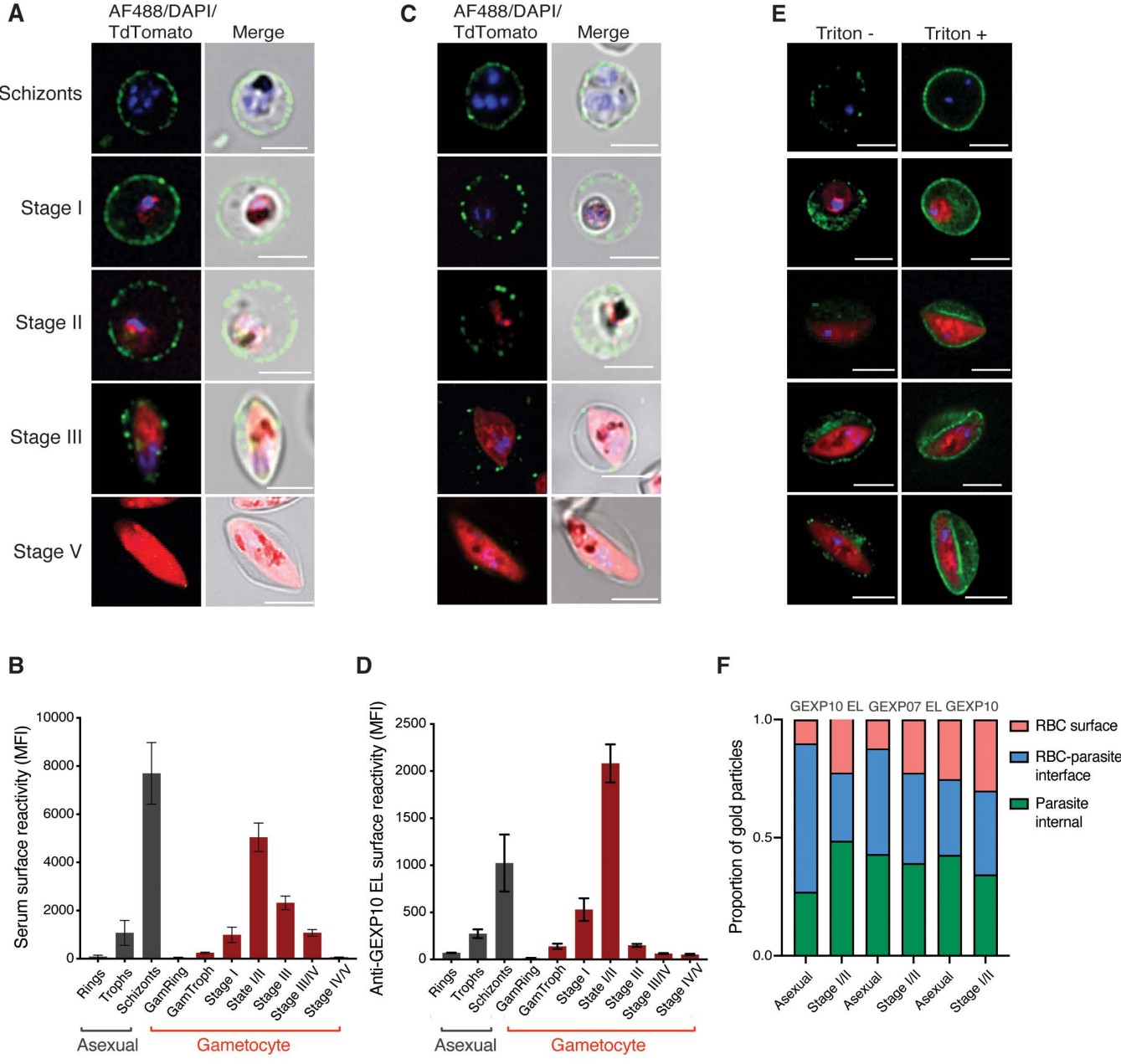

**Fig 1. Dynamics of surface antigen exposure in *P. falciparum* gametocytes.** A,B. Human serum reactivity across asexual and gametocyte development by live microscopy and flow cytometry detection in Pf2004 parasites. (A) Representative images of asexual and gametocytes after live labelling and fluorescence microscopy using immune serum, and (B) flow cytometry quantification of the same samples. C,D. Reactivity of GEXP10 EL antibodies in live cells by (C) fluorescence microscopy and (D) flow cytometry. E. Human serum reactivity in PFA fixed (left) and PFA fixed and Triton X-100 permeabilized cells (right) in asexual and gametocyte stages. F. Quantification of antigen localization based on immuno EM. Asexual stages: 24-36 hpi, GI-IIa: D4, GIIb-III: D6, GIII-IV: D8, GIV-V: D11. Blue: DAPI, green: IgG, red: Tdtomato. B, and D represent the mean from three biological replicates. F, counting a minimum of 30 images per stage and antibody.

antigens expressed both in asexual and gametocyte stages, PfGEXP07 and PfGEXP10 [12,16] and observed that reactivity was limited to early developmental stages (Fig 1C, 1D and S1). The on and off pattern of giRBC immune recognition was in agreement with our previous observations [12] and confirmed in the reference parasite line NF54 (S1 Fig). Interestingly, permeabilization experiments revealed that most of the serum and GEXP10 signal remained within the iRBC and inside the parasite, even at peak recognition in stage II gametocytes. In contrast, only a small proportion is expressed on the iRBC surface (Figs 1E, F and S2). This is similar to observations in asexual stages, where only a fraction of the major surface protein, PfEMP1, is delivered to the iRBC surface while the majority remains within the host cell [21]. Quantification of antigen distribution based on immunoelectron microscopy using GEXP10 and GEXP07 antibodies confirmed our flow cytometry and immune fluorescence analysis, showing that in mature asexual stages and immature gametocytes a majority of antigen remains inside the RBC (Figs 1F and S2). Interestingly, a major pool of antigen also accumulated internally in mature gametocytes, however the role of these antigens in gametocytes or the resulting gametes remains to be determined. To ensure that serum and antibody recognition of live stage II gametocytes was indeed limited to iRBC surface antigens (as opposed to RBC internal antigens due to leaky membranes), we measured the membrane integrity of live cells using an antibody against human alpha spectrin, an internal RBC protein. We observed little to no binding of the antibody to live cells, while fixed and permeabilised cells showed high levels of binding. These experiments demonstrate that our data with live cells represent serum and specific antibody reactivity on the iRBC surface only, while no significant signal is derived from RBC internal antibody binding (S3A Fig). Altogether these data confirmed and extended our previous observations (Dantzler et al, 2019) [12], demonstrating reversible iRBC surface exposure of parasite antigens peaking at stage II gametocytes while internal antigen pools change only minimally during gametocyte development.

**Reversible PS flipping tracks with antigen exposure dynamics**

In asexual parasites, antigen surface exposure peaks in schizonts, i.e., right before the asexual cycle ends. In contrast, the peak of antigen surface exposure in gametocytes is at stage II and hence must be reversible. This reversible dynamic during gametocyte development may be the result of major membrane remodelling to remove parasite antigens from the iRBC surface. To test this hypothesis, we quantified phosphatidylserine (PS) surface exposure during gametocyte development. Healthy RBCs maintain membrane lipid asymmetry between the inner and outer leaflet: typically phosphatidylcholine (PC) and sphingomyelin (Sph) are located on the outer surface, while phosphatidylethanolamine (PE) and phosphatidylserine (PS) are located exclusively internally [22]. Aged and damaged RBCs undergo eryptosis where PS is expressed on the outer surface, signalling macrophages to internalise and remove the damaged cells from circulation. Similarly, iRBCs expose PS on the outer leaflet of the host cell membrane in mature asexual stages [18], because the parasite disrupts the maintenance of membrane asymmetry during schizogony (reviewed in [23]). To investigate whether PS becomes exposed on the iRBC surface during gametocyte development, we carried out a time course experiment and quantified Annexin V binding to surface exposed PS on intact iRBCs and uninfected RBC (uRBC) controls. Annexin V binding started in early gametocyte development, peaking at stage II and waning in mature stages with lowest PS exposure in stage V (Fig 2). This pattern was observed in both the Pf2004 and NF54 parasite lines by live fluorescence imaging and flow cytometry (Fig 2A–D). In NF54, we observed a much steeper decline in PS exposure at stage IV-V, possibly due to the difference in stage IV-V maturation between the two parasites lines (Fig 2C). We observed no annexin V binding to uRBCs across experiments, demonstrating that the observed annexin V binding to iRBCs is not a result of old senescent (and hence PS positive) RBCs (S3C Fig). Therefore, the observed binding to aiRBCs and immature giRBCs is the result of the parasite modifying the membrane lipid symmetry of their host cells. Using a specific PS antibody, we noted little change in internal PS staining, in contrast to the reversible surface dynamics (Fig 2E). Altogether these data revealed a reversible exposure of PS to the iRBC surface during gametocyte development. Importantly, our observations suggest that

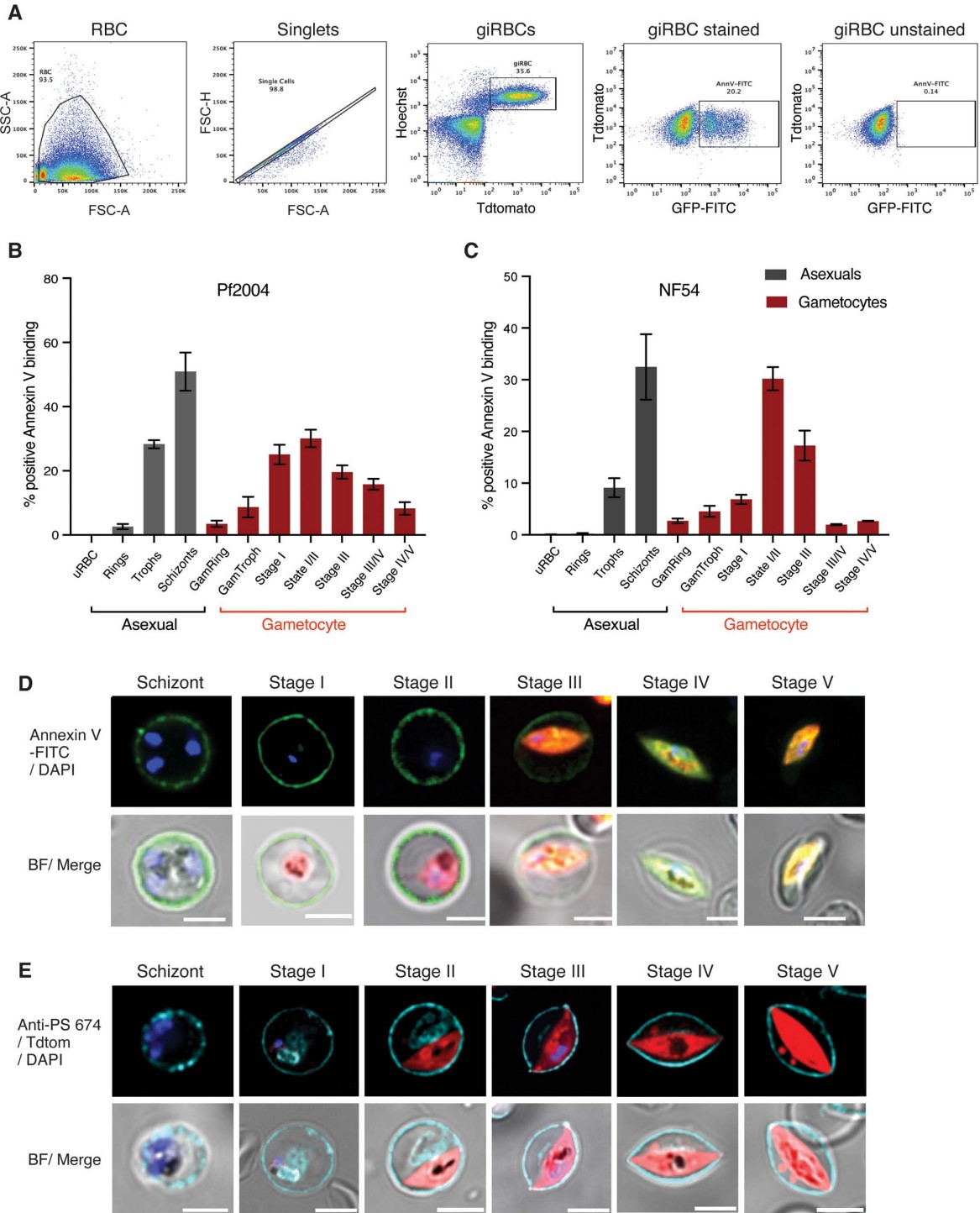

**Fig 2. Reversible membrane remodelling tracks with antigen exposure during gametocyte development.** A. Example gating strategy for MACS-enriched stage II gametocytes showing annexin V labelled cells in comparison to unstained controls. B-C. Annexin V labelling in live MACS-enriched asexual and gametocyte stages by flow cytometry (B: Pf2004, C: NF54) and fluorescence microscopy (D: Pf2004). E. PS staining using a monoclonal anti-PS antibody in fixed and permeabilized asexual and gametocyte stages (Pf2004). Blue: DAPI, green: Annexin V-FITC, red: Tdtomato, Magenta: anti-PS antibody-647. B, C represent the mean from three biological replicates.

parasite surface antigen exposure and iRBC membrane remodelling coincide during early gametocyte development before both processes are reversed at later stages. Notably, both annexin V and antibody binding to the iRBC surface is subject to significant RBC batch variation (S3B and S3C Fig).

## Blocking antigen surface expression also affects PS flipping

The similar dynamics of reversible surface antigen exposure and PS flipping also suggested that the two processes may be interlinked. This hypothesis was supported by the strong positive correlation between surface antigen exposure and PS flipping across individual samples (Fig 3A). To directly test this hypothesis, we used specific perturbations targeting each process individually. The *P. falciparum* protein export pathway, including processing of exported proteins by the aspartyl protease, Plasmepsin V (PMV) (reviewed in [24]) is operational in asexual and gametocyte stages [25]. To determine whether PMV affects iRBC surface expression during gametocyte development we utilized an inhibitor of PMV (WEHI 842) [26,27] in the background of the Pf2004/Tdtom-Gexp02/GFP dual reporter line. When added at gametocyte ring stage (10–16 hpi) the PMV inhibitor affected early gametocyte development in a concentration dependent manner (Fig 3B, C), as previously reported [25]. Using the Tdtom reporter signal intensity as a proxy for stage development, we noted that a minor proportion of gametocytes was not arrested and instead developed normally at lower drug concentrations (1 and 2 µM) (Fig 3D). Gating for this subset of developing gametocytes, we then used the Gexp02/GFP reporter to test whether the PMV inhibitor blocked GEXP02 export. Further experiments using the same gating strategy and 2 µM of drug revealed a significant reduction in serum and GEXP10 antibody surface binding in asexual parasites (Fig 3E) and immature gametocytes (Fig 3F and 3G). Conversely there was no effect on gametocyte development or GEXP10 export when the drug was added at stage II and later (Fig 3H). However, we observed a significant reduction in serum surface labelling of mature gametocytes (Fig 3I), suggesting that some PMV dependent antigen export continues during gametocyte maturation.

Our observation of similar dynamics in surface antigen expression and PS surface exposure suggested shared underlying mechanisms. To test whether these two processes are interlinked, we investigated whether membrane remodelling is dependent on parasite antigen export. Specifically, we tested the effect of PMV inhibition on PS exposure during gametocyte development. Indeed, adding 2 µM of PMV inhibitor at gametocyte ring stages resulted in a significant reduction in PS exposure at stage II gametocytes (Fig 3J). In contrast, when PMV inhibitor was added at stage II gametocytes, PS surface exposure was not affected in late gametocyte development (Fig 3K). Altogether these data suggest that PS flipping in early gametocytes at least partially depends on an exported parasite activity, while the reverse process during later gametocyte maturation does not.

## PS exposure and reversal can be blocked by scramblase inhibition

In a next series of experiments, we investigated the possible mechanisms underlying the reversible PS flipping on the iRBC surface. Recently, a *P. falciparum* phospholipid scramblase (*Pf*PL-SCR1, accession number PF3D7_1022700) with expression in late asexual stages and gametocytes was reported. Phospholipid scramblases are enzymes that translocate phospholipids across membranes in an ATP-dependent fashion. A *Pf*PL-SCR1-GFP fusion protein localized to parasite internal membranes both in asexual and gametocyte stages (Fig 4A), suggesting a role in PS scrambling in the parasite [28]. Genetic deletion did not have an effect on parasite growth; however, PS exposure was not investigated in the original study. We hypothesized that this parasite scramblase could be involved in iRBC membrane remodelling. Deletion of the parasite scramblase in a *Pf*PL-SCR1 KO line resulted in significant reduction of PS exposure in late asexual stages and early gametocytes stages, while loss of surface PS in later stages was unaffected (Fig 4B–F). Interestingly, genetic deletion of the scramblase did not have an effect on surface antigen exposure (Fig 4G). Altogether these data demonstrate that the parasite scramblase is required for efficient PS flipping on the iRBC surface in asexual and early gametocyte stages, but not for the reversal during later gametocyte development.

To further investigate PS flipping on the iRBC surface during gametocyte development, we tested four mutually compatible scenarios: i) parasite-induced channels could cause a calcium influx affecting cell membrane scrambling and PS

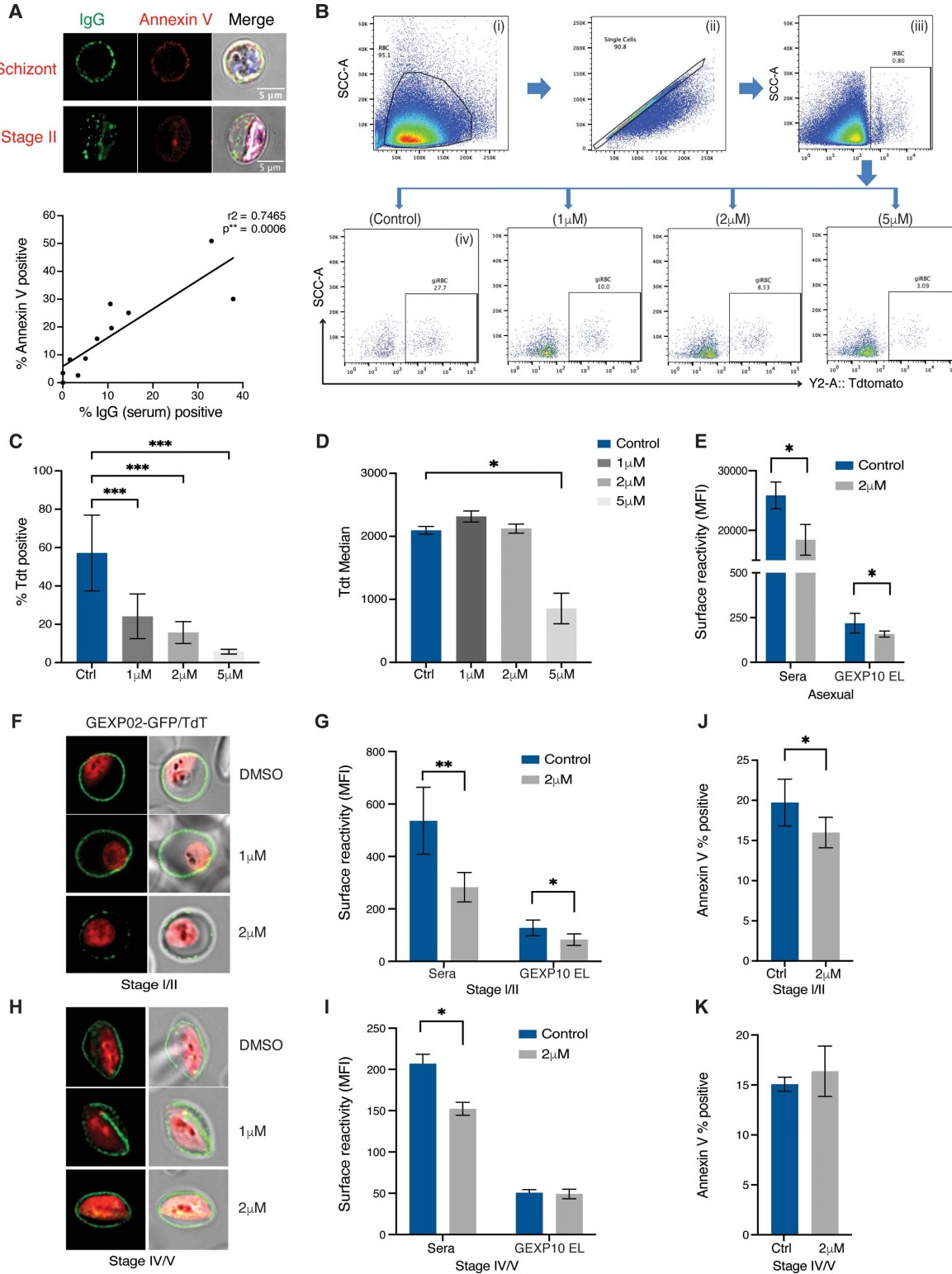

**Fig 3. Inhibition of antigen export blocks gametocyte development and surface recognition.** A. Co-staining of surface labelling with human serum and PS with annexin V. Representative images of asexual schizont (top) and stage II gametocytes (bottom). Graph showing correlation of surface labelling and annexin V staining across individual samples (bottom panel). Each point represents the mean of three biological replicates of the combined data from the serum (S1G Fig) and PS exposure (Fig 2A) time course experiments. For each experiment a minimum of 10,000 cells was acquired. B.

Flow cytometry data gating strategy. Shown is an example for immature stage II gametocytes, day 5 post induction. Cell population was gated to remove debris (i), followed by single event gating to remove doublets (ii). iRBCs were gated on nuclear dye (iii) and gametocytes on Tdtomato (iv). The Tdtomato % positive cells (gametocytemia) varied based on PMV inhibitor concentration. Left to right: DMSO control, 1µM, 2µM and 5µM PMV inhibitor. C,D. PMV inhibitor arrests early gam development at higher doses based on Tdt MFI quantification. C. Quantification of gametocytemia in MACS enriched parasites by Tdtomato positivity across three PMV inhibitor concentrations. D. Quantification of Tdtomato median fluorescence in Tdtomato positive cells gated in C. E. PMV inhibitor at lower doses blocks antigen export in mature asexual stages (36-44hpi). F. GEXP02 and TdTomato fluorescence upon PMV inhibitor incubation. PMV inhibitor at lower doses blocks antigen export in immature gametocytes. Shown are representative images of immature gametocytes (Stage II). G. Flow cytometry quantification of immune reactivity (human serum and GEXP10 EL) upon PMV inhibitor incubation in immature gametocytes. H-I. PMV inhibitor has less effect on antigen export in mature gametocytes. H. GEXP02 and TdTomato fluorescence upon PMV inhibitor incubation. Shown are representative images of mature gametocytes (Stage IV-V). I. Quantification of immune reactivity (human serum and GEXP10 EL) (I). J, K. Annexin V staining upon PMV inhibitor incubation in immature gametocytes (J) and mature (K) gametocytes. Blue: DAPI, green: IgG (A)/ GEXP02-GFP (F,H), red: Tdtomato. C,D,E,G,I,J,K represent the mean from three biological replicates. Statistics represent paired t-test on inhibitor treated *vs* untreated (DMSO only) control. *p*-value <0.0001 ****, *p*-value <0.001 ***, *p*-value ≤ 0.01 **, *p*-value ≤ 0.05 * and *p*-value > 0.05 ns.

exposure (i.e., host flippase or scramblase activity [18]); ii) high intracellular calcium levels could activate the Gardos channel (a host potassium channel) leading to KCl and water efflux and increased PS exposure [29]); iii) increased cellular calcium could lead to translocation of host protein kinase Ca (PKCa) to the plasma membrane and further increase of calcium uptake (via host PKC and Syk kinase) [30] – a positive feedback mechanism inducing PS exposure; iv) inhibition of ATP hydrolysis could result in increased intracellular calcium and PS exposure, as has been shown in asexual iRBCs [18]. We used a series of commercially available inhibitors to test each of these scenarios individually: (i) flippase inhibitor (N-ethylmaleimide, NEM) [31] and scramblase inhibitor (Ethaninidothioic acid, R5421) [32], ii) Gardos channel blocker (Senicapoc) [29], iii) PKCa inhibitor (Chelerythrine chloride, CHE) and tyrosine kinase (Syk) inhibitor [33,34], and iv) ATP hydrolysis inhibitor (Vanadate) [18]. To exclude a possible effect on parasite viability we tested each inhibitor in asexual parasites first, with the aim to select the highest non-toxic concentration (i.e., no effect on parasite multiplication rate, PMR) for subsequent assays in gametocytes (S4 Fig). Titration series were set up, starting with concentrations from the studies referenced above. Parasites were treated at trophozoite stage and PS exposure by annexin V binding quantified at the mature schizont stage. PMR was quantified after one cycle of reinvasion at the next trophozoite stage (48hrs drug exposure). In asexual parasites, increasing concentrations of Syk and PKCa inhibitors reduced both surface PS exposure and PMR. In contrast, scramblase, flippase and Gardos channel inhibitors had no significant effect on PS exposure nor PMR in asexual parasites. Finally, we were not able to determine a non-toxic concentration for the ATP hydrolysis inhibitor (i.e., all concentrations tested significantly reduced parasite growth) (S4E Fig). Based on these experiments we selected drug concentrations for single dose treatment in gametocytes. Specifically, for Syk and PKCa inhibitors we used the highest non-toxic concentration of 2.5 µM (i.e., no significant effect on PMR). For the flippase inhibitor we used 20 µM, for the scramblase inhibitor 100 µM, and for the Gardos channel inhibitor we used the published IC50 concentration for *P. falciparum* of 6.7 µM. For vanadate we used 1.25 µM, the highest concentration that allowed some gametocyte development (all higher concentrations completely blocked growth). To measure an effect on PS surface exposure, inhibitor was added at early stage I gametocytes (3 days post induction) for 48 hours and PS surface exposure quantified at stage II gametocytes (5 days post induction). To measure an effect on surface PS internalisation, inhibitors were added at stage III gametocytes (6 days post induction) and surface PS measured at stage IV-V gametocytes. None of the inhibitors had any significant effect on PS surface flipping in stage II gametocytes (Fig 5A). In contrast, scramblase, flippase and ATP hydrolysis inhibitors significantly reduced PS internalisation in mature gametocytes (Fig 5B). However, both flippase and ATP hydrolysis inhibitors also blocked gametocyte development (Fig 5C), thus the observed effect on PS internalisation is likely the result of arrested or dying gametocytes. In conclusion, the only inhibitor that specifically affected PS surface internalisation was the non-specific scramblase inhibitor, R5421. This inhibition was specific to PS, while antigen surface expression or immune recognition of the iRBC surface remained unaffected (Fig 5D). Altogether these data suggest that loss of PS exposure during gametocyte maturation depends on a host scramblase, and that process is independent of the loss of surface antigens.

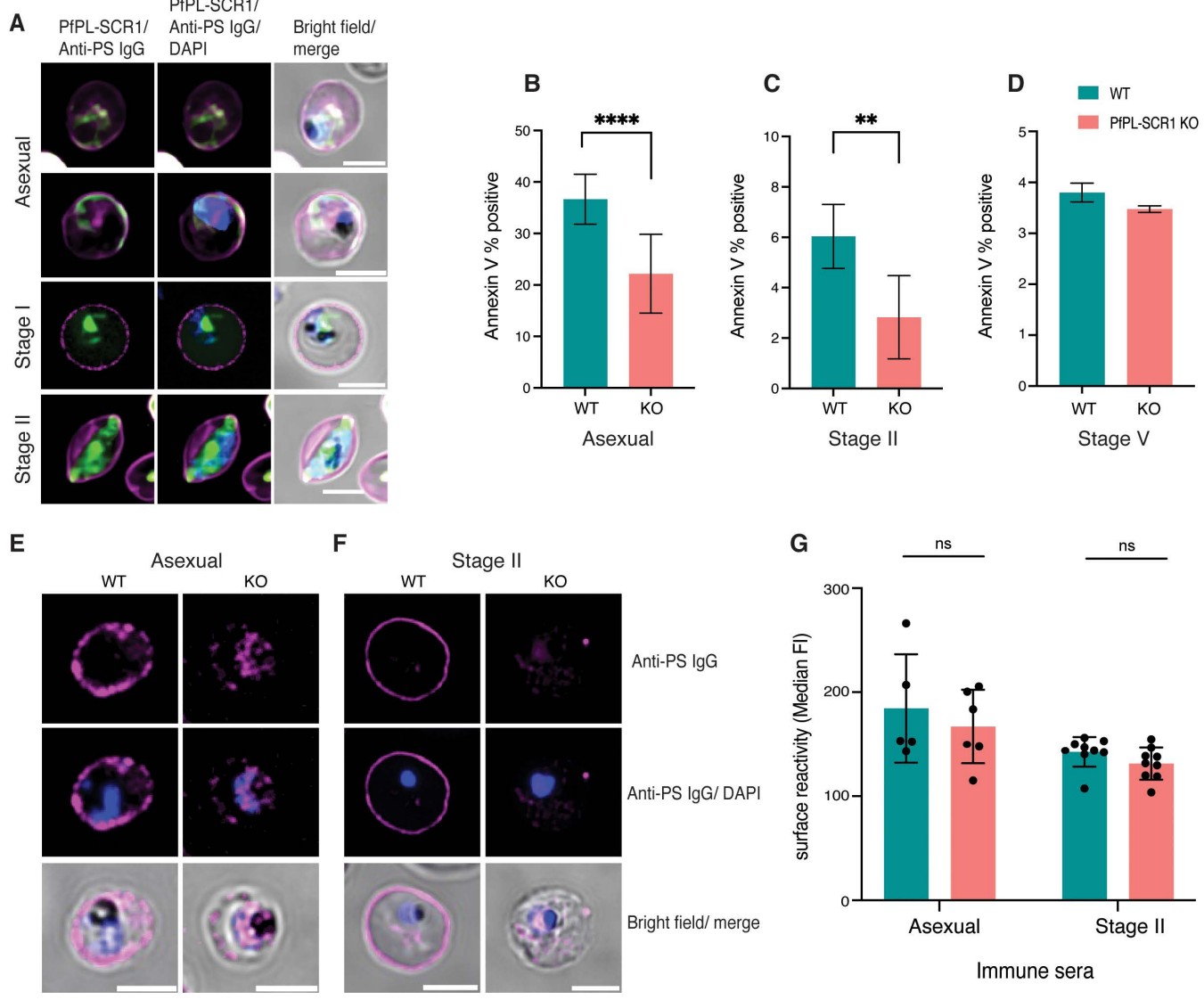

**Fig 4. A *P. falciparum* scramblase affects PS surface exposure.** A. Localisation of *Pf*PL-SCR1-GFP in asexual parasites and immature gametocytes (stage I and stage II) in relation to surface PS. B-D. Effect of *Pf*PL-SCR1 KO on PS exposure in schizonts (B), immature (C) and mature (D) gametocyte stages. E,F. Representative images of live PS staining in *Pf*PL-SCR1 KO parasites using anti-PS antibodies. Shown are asexual schizont stages (E) and Stage II gametocytes (F). G. Serum recognition in the *Pf*PL-SCR1 KO measured by flow cytometry. Blue: DAPI, green: *Pf*PL-SCR1-GFP, Magenta: anti-PS antibody-647. B, C, D, G represent the mean from three biological replicates. Statistics represent paired t-test on *Pf*PL-SCR1 KO *vs* parental WT line. *p*-value <0.0001 ****, *p*-value <0.001 ***, *p*-value≤ 0.01 **, *p*-value ≤ 0.05 * and *p*-value > 0.05 ns.

## Host cell remodelling impacts parasite clearance mechanisms by immune cells

We have previously demonstrated that immune sera from malaria patients can induce opsonic phagocytosis of gametocyte infected RBCs, suggesting that surface antigens elicit these immune responses. To further investigate the effect of surface antigen expression on gametocyte infected RBC clearance, a potential host immune mechanism of transmission reducing immunity, we measured antibody dependent opsonic phagocytosis using commercial THP-1 cells [17], across gametocyte maturation. We observed significant phagocytosis of asexual and immature gametocyte, but not mature

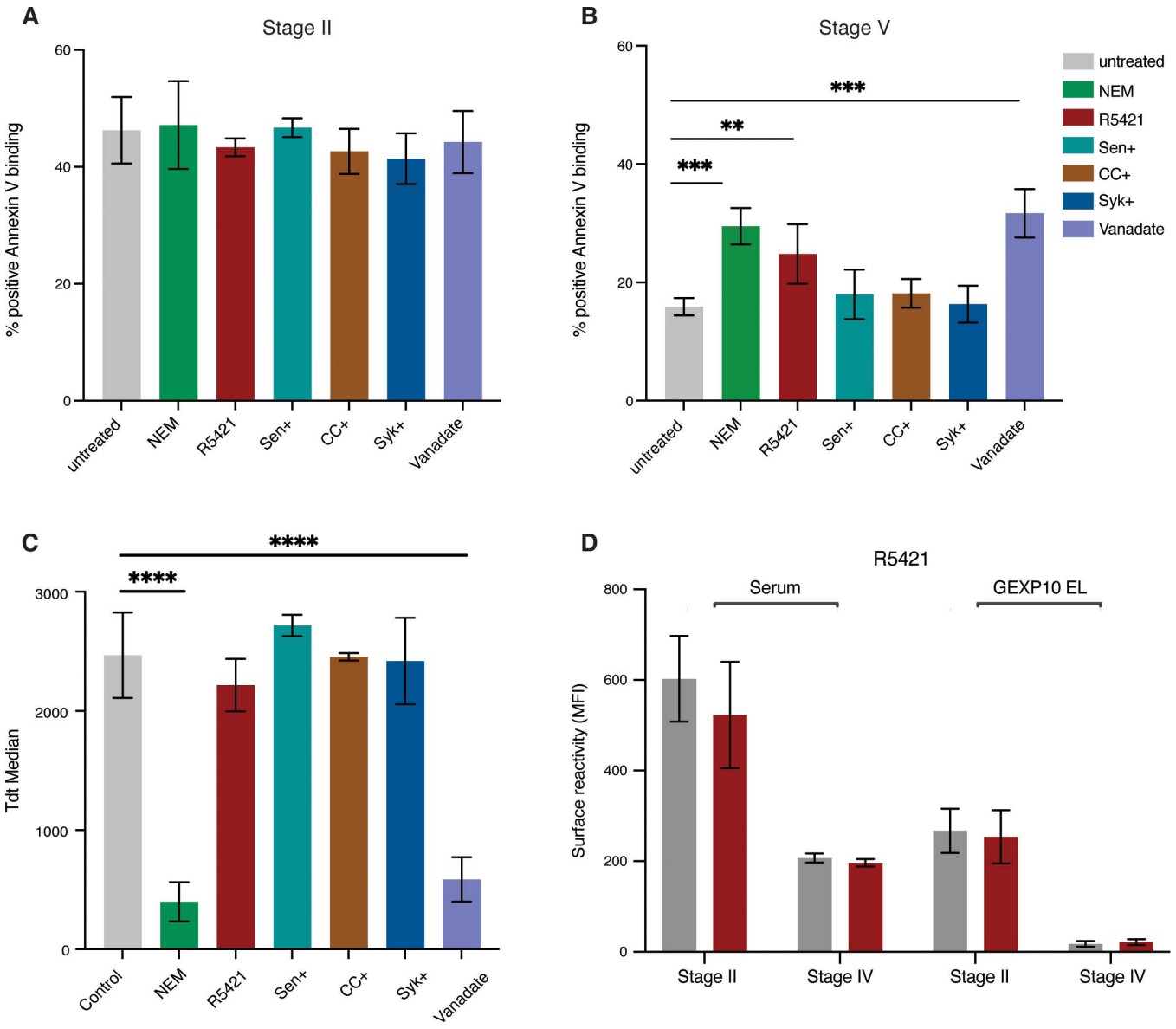

**Fig 5. Blocking PS flipping in gametocytes using scramblase inhibitor.** A. Single dose incubation of compounds tested for effect on immature gametocytes (Stage I-II). Concentration of compounds: NEM (20µM), R5421 (100µM), Senicapoc (6.7µM), CHE (2.5µM), Syk (2.5µM), Vanadate (1µM). B. Single dose of compounds tested for effect on mature gametocytes (Stage IV-V). Concentrations for all compounds as in A. C. Tdtomato reporter median fluorescence in mature gametocytes (Stage IV-V) to determine compound effect on gametocyte maturation. D. Effect of R5421 on human serum and GEXP10 EL antibody surface recognition in immature and mature gametocytes. A-D represent the mean from three biological replicates. Statistics on B-C represent paired t-test on inhibitor treated *vs* untreated (DMSO only) control. *p*-value <0.0001 ****, *p*-value <0.001 ***, *p*-value ≤ 0.01 **, *p*-value ≤ 0.05 * and *p*-value > 0.05 ns.

gametocytes, when opsonized with immune sera (S5A Fig). Uptake of opsonized mature gametocytes was comparable to background phagocytosis observed when opsonized with non-immune sera.

Addition of Plasmepsin V inhibitor in these experiments reduced surface antigen exposure in asexual and immature gametocyte stages (S5B and S5C Fig) but did not result in reduced phagocytosis of either stage (Fig 6A and 6B). A recent study demonstrated that PS exposure leads to increased asexual iRBC uptake by monocytes [18]. Here we tested

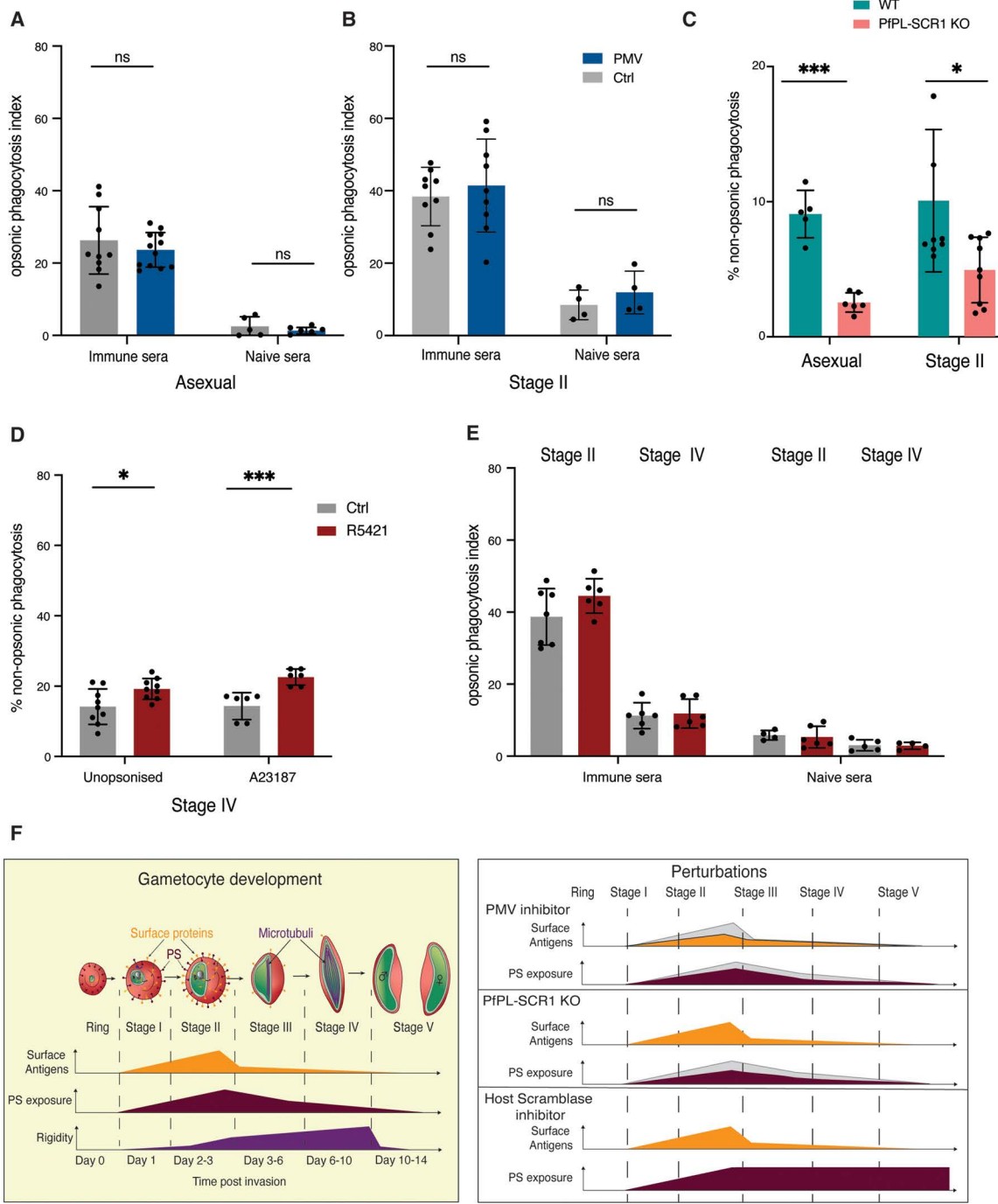

**Fig 6. Phagocytosis of iRBCs upon perturbations.** A-B. Effect of PMV inhibitor on phagocytosis of asexual (A) and immature gametocyte (B) stages using immune serum vs US control serum for opsonisation. C. Effect of *Pf*PL-SCR1 KO PS exposure on iRBC phagocytosis in absence of serum (no opsonisation). Shown are asexual schizonts and ring stages. D. Effect of R5421 on Stage IV-V iRBC non-opsonic phagocytosis in absence of serum and upon addition of the calcium ionophore A23187. E. Effect of R5421 on Stage IV-V iRBC opsonic phagocytosis in presence of serum. F. A model of the remodelling process during gametocyte maturation and effect of the various perturbations. A-E represent the mean from three biological replicates. For the opsonic phagocytosis data, phagocytosis index is determined by subtracting average negative control %DHE and dividing by average positive control %DHE. For the non-opsonised phagocytosis data, % non-opsonic phagocytosis indicates the proportion of DHE positive THP-1 cells (i.e., the proportion of iRBC uptake).

whether the reduced surface PS levels in the *Pf*PL-SCR1 KO line had any effect on asexual iRBC uptake by THP-1 cells. Indeed, we observed significantly decreased non-opsonic (i.e., serum-independent) phagocytosis of both asexual and stage II gametocytes of the KO compared to wild type parasites (Fig 6C).

Finally, we determined whether blocking host scramblase activity in mature gametocytes impacted phagocytic clearance. We used the scramblase inhibitor R5421 to treat stage II gametocytes and block PS flipping off the iRBC surface (see Fig 5) and measured the effect on non-opsonic phagocytosis. This experiment revealed a significant increase in phagocytic uptake of treated mature gametocytes, similar to the effect in our positive control (Fig 6D). The observed effect was specifically due to surface PS because there was no effect on antibody-dependent opsonic phagocytosis (Fig 6E). Altogether these data suggest that phagocytosis of immature gametocytes is at least partially driven by PS exposure, and that blocking PS flipping has potential to negatively impact gametocyte survival and thus potential to reduce transmission.

## Discussion

*P. falciparum* has an unusually long gametocyte development compared to other *Plasmodium* species, during which the infected cell undergoes an extensive morphological transformation. Parasites export antigens onto the surface of host iRBCs during early gametocyte development in a similar trajectory as in the asexual stages. In both stages there is a gradual increase in surface antigen export starting around 20 hrs post invasion (late ring stages) and peaking in schizonts and stage II gametocytes, respectively. While schizonts burst to release invasive daughter cells with a new set of surface antigens required for RBC invasion, antigens are actively removed from the iRBC surface before maturation in Stage V gametocytes.

The data from the current study corroborate our earlier observations that sampled part of the gametocyte maturation cycle [12]. We further demonstrate that the loss of antigens is restricted to the iRBC surface as internal pools do not decline with gametocyte maturation (S2 Fig). The purpose of these internal antigen pools remains unclear, but we hypothesise that antigen export to the iRBC surface is tightly regulated to minimise antigen exposure to the host immune system. The internal antigen pool may also serve additional structural roles within the host RBC, as has been demonstrated for GEXP07 [35] and for STEVOR antigens [36]. Our study also confirms that gametocytes rely on protein export via Plasmepsin V processing for surface antigen expression. Inhibition of PMV resulted in reduced stage II gametocyte surface recognition by serum and by antibodies to candidate gametocyte antigens GEXP07 and GEXP10. The incomplete block of surface recognition may be due to antigen export before PMV inhibitor treatment and/or the inhibitor concentration used. It is worth noting that at higher PMV inhibitor concentrations gametocyte development was greatly impeded (Fig 3) [25]. The mechanism of antigen removal from the iRBC surface remains to be determined. There are at least three possible scenarios. First, membrane-bound antigens (and accessory proteins associated with those) could be cleaved by a family of intramembrane serine proteases, the rhomboid proteases. The *P. falciparum* genome encodes for eight rhomboid proteases, some of them are required for processing of parasite ligands essential for RBC invasion [37]. However, none of the parasite rhomboid proteases are known to be exported into the host cell and RBCs do not contain rhomboid proteases either. Second, parasite surface antigens could be internalised via membrane flipping. However, our experiments have demonstrated that loss of parasite surface antigens is independent of PS internalisation (which depends on a host scramblase). Third, parasite surface antigens could be shed with the release of extracellular vesicles (EVs). Indeed, such small vesicles are released during RBC maturation and their numbers increase significantly in aiRBCs [38]. EVs derived from aiRBCs also contain parasite surface antigens [38], demonstrating their shedding through this mechanism. It is therefore possible that the selective shedding of parasite surface antigens during gametocyte maturation is also facilitated via EV release.

Importantly, our experiments reveal a dynamic change in lipid asymmetry during gametocyte maturation: PS levels on the outer iRBC leaflet peak in immature stages, before they decrease drastically during final gametocyte maturation. Uninfected RBCs and other mammalian cells generally maintain membrane lipid asymmetry with the anionic phospholipids PS

and PE kept in the internal side facing the cytoplasm, while PC and SM are exposed externally to the cell environment. In homeostasis this heterogenous lipid distribution is maintained by three membrane integral phospholipid translocases, termed flippases, floppases, and scramblases [39]. Flippases are ATP-dependent and unidirectionally shunt PS and PE to the inner lumen of the membrane [40]. Floppases are ABC-transporters that mediate the movement of phospholipids in the reverse direction [41] while scramblases are ATP-independent and act to randomize lipid distribution by bidirectionally translocating lipids between leaflets [42]. Disruption of this asymmetry signals cell damage and apoptosis, resulting in removal of the damaged cell. The hallmark of apoptosis is constitutive and non-reversible PS translocation to the outer surface of the cell membrane, which signals immune cells to clear the damaged cell. In the case of RBCs, the normal ageing process results in PS exposure and removal by splenic macrophages. Similarly, eryptosis due to RBC damage by oxidative or other cellular stresses involves PS exposure and subsequent phagocytosis of the damaged cell. It has been hypothesised that *Plasmodium* (and a variety of other parasites) has adopted a strategy of non-classical apoptotic mimicry [43], where parasites induce apoptosis and co-opt the exposed PS of the apoptotic cell for its own survival, for instance by increasing the cytoadhesive capacity of the iRBC. PS exposure peaks in mature asexual stages and is important for merozoite egress [18]. Earlier work suggested that PS exposure on iRBCs facilitates binding to endothelial cells via interaction with CD36 and CXCCL16 [44,45]. Here we demonstrate that immature gametocytes expose PS on the iRBC surface, and that this exposure is reversed in mature gametocytes. The role of this remodelling is still unclear. We propose a role in cytoadhesion that supports retention of the rigid immature gametocyte in the extravascular environment. Cytoadhesive antigens remain elusive in gametocytes and therefore PS could play a more central role in mediating sequestration in the extravascular space of the bone marrow. Sexual stage parasites are generally considerably lower in abundance than their asexual counterparts and take longer to develop and mature in the host, and therefore they might require additional measures for immune evasion. The coincidental loss of PS and surface antigens is both a novel and logical observation, as it allows circulation of mature gametocytes without eliciting immune clearance responses.

PS exposure in immature gametocytes depends on a parasite phospholipid scramblase, *Pf*PL-SCR1, that localises to internal parasite membranes while exerting an effect on lipid asymmetry on the host cell membrane. Scramblases have been identified at various internal membranes, including endoplasmic reticulum (ER), nuclear envelope and mitochondria and they can exert their function both in *cis* (i.e., in the same membrane) or in *trans*. For example, the mammalian scramblase TMEM16K is localised at the ER membrane and regulates PS both in the ER and in organellar membranes of the endolysosomal system [46]. Our data suggest that exported proteins facilitate the effect of PfPL-SCR1 at the iRBC membrane, as supported by the sensitivity of PS exposure to PMV inhibitor treatment. In addition, recent studies have demonstrated that PMV processes non-exported proteins and hence inhibition of this activity may also directly or indirectly affect *Pf*PL-SCR1 function [47,48]. The mechanism of PS exposure by *Pf*PL-SCR1 is potentially similar in immature gametocytes and mature asexual stages [18], while the removal in mature gametocytes occurs via an unknown mechanism clearly independent of *Pf*PL-SCR1. To investigate the mechanism of PS removal, we tested a series of compounds blocking possible modes of action including host flippases, scramblases, calcium channels and kinase signalling. None of the compounds affected either asexual or immature gametocyte PS exposure. In contrast, a scramblase inhibitor blocked PS removal in mature gametocytes (without affecting gametocyte development), suggesting that PS dynamics in maturing gametocytes are regulated by scramblase activity. Altogether, our observations that a combination of host and parasite scramblases is required for membrane lipid remodelling in gametocytes provide novel insights into the intricate mechanisms of host-parasite interactions in *Plasmodium* (Fig 6F).

Our study has some limitations. First, we have noted that only a subpopulation of 15–40% of iRBCs are positive for IgG binding and Annexin V staining, even at peak time (i.e., Stage II gametocytes). We have previously observed such limited IgG reactivity [12] and limited PS reactivity has been reported by others [18]. Our data also suggest that the variation in IgG binding and PS staining across experiments is largely driven by differences across RBC donor batches, as has previously been shown for human IgG reactivity to iRBCs [49]. Technical limitations in the detection of fluorescence signal on the

surface of live cells and the relatively low abundance of parasite antigens on the iRBC surface contribute to low MFI and % positive rates in our experiments. We hypothesize that PS and antigen surface exposure are essential for the survival of immature gametocytes. Therefore, we assume that the variable and low percentage of iRBCs with PS exposure and IgG reactivity is an artifact of parasite *in vitro* culture. Second, we did not observe an effect of the PMV inhibitor on phagocytosis of asexual stages and immature gametocytes. However, the inhibitor efficiently blocks antigen surface exposure, and the level of surface exposure is strongly correlated with the level of phagocytosis [12]. It is therefore likely that a putative effect of PMV inhibitor on iRBC phagocytosis is masked by an unknown secondary effect that triggers phagocytosis independent of PEXEL-dependent parasite antigen exposure. The PMV inhibitor is essentially a PEXEL motif peptide that outcompetes the natural substrate. It is highly specific, and no other protease has been shown to cleave PEXEL peptides [27]. Therefore, an off-target effect is unlikely. As blocking PEXEL cleavage accumulates substrate antigens in the ER and jams up the secretory pathway, it is most likely that the PMV inhibitor induced a stress response that altered the iRBC surface.

Developing anti-gametocyte approaches is key to eliminating malaria. Gametocytes are the only transmissible stages and blocking this stage will play a significant role in elimination strategies, especially when transmission has been brought down by other preventative public health measures like intensive vector control. Whilst transmission-blocking interventions have traditionally focused on blocking parasite development inside mosquitoes, anti-gametocyte interventions can be equally potent and have the operational advantage that their effectiveness can be confirmed by monitoring gametocyte carriage in human populations. Indeed, our phagocytosis data provide a proof-of-concept for the development of novel intervention strategies that either inhibit gametocyte sequestration by blocking antigen and/or PS exposure, or by inducing splenic clearance of circulating gametocytes by blocking PS and/or antigen removal. These data suggest that targeting the host RBC scramblase can block parasite transmission by increasing splenic clearance, offering a new approach for host targeted therapies in the fight against malaria. Altogether, our data provide a foundation for further mechanistic studies and targeted approaches for novel transmission blocking therapeutics.

## Materials and methods

### Ethics statement

The project was approved by IRB boards of Harvard T.H. Chan School of Public Health, University of Malawi College of Medicine, Michigan State University, and University of Maryland. Tanzanian immune sera were collected as described [50].

### Immune serum samples and ethical approval

Archived sera samples from Malawi and Tanzania were used as highly reactive pools for immune assays. Sera from Malawi were collected as previously described [12], from 3 district hospitals (Ndirande, Thyolo, Chikhwawa) in the region of Blantyre, Malawi. Samples were collected by staff from the International Center of Excellence for Malaria Research (ICEMR) surveillance study after written informed consent was obtained from participants and/or their parent(s) or guardian(s). Naïve sera controls were obtained from Interstate Blood Bank (Memphis, TN, USA) or Radboud University Medical Centre (Nijmegen, the Netherlands).

### Transgenic parasites generation

Generation of GEXP02 (PlasmoDB: PF3D7_1102500) GFP-tagged parasite lines: To generate Pf2004/0164Tdt line the pB_gC Cas9/sgRNA plasmid (ref) was modified to generate pBgC-GEXP02 as follows. Two complementary oligonucleotides (G02GFP_gRNA_fwd and G02GFP_gRNA_rev) encoding the sgRNA target sequence for GEXP02 and appropriate single-stranded overhangs were annealed and inserted into the sgRNA expression cassette using *Bsa*I-digested pB_gC and In-Fusion cloning. The donor cassette was generated using In-Fusion HD cloning kit (Takara) to join PCR fragments

GEXP02 homology region (HR) 1 amplified from Pf2004_164Tdtom gDNA using primers G02GFP_HR1_fwd and G02GFP_HR1_rev, and GEXP02 HR2 generated using primers G02GFP_HR2_fwd and G02GFP_HR2_rev. PCR was performed with Phusion HF DNA polymerase (NEB) with the following thermocycling conditions: initial denaturation at 95°C for 2 min, thermocycling with denaturation at 95°C for 30 sec, annealing gradient ramp from at 45°C to 55°C for 30 sec, elongation at 61°C for 1 min for 8 cycles, 22 additional cycles [denaturation at 95°C for 30 sec, annealing at 55°C for 30 sec, elongation at 61°C for 1 min] and final elongation step at 61°C for 8 min. In the edited strain (Pf2004/164tdTom –GEXP02-GFP), integration of *bsd* resistance marker was verified by PCR using primers G02KO_INT_fwd and BSD_5'UTR_R primer (testing 5' integration) and BSD_3'UTR_F and 3'INT G02 GFP rev (testing 3' integration). PCR conditions were as follows: initial denaturation at 95°C for 2 min, thermocycling with denaturation at 95°C for 30 sec, annealing at 48°C for 30 sec, and elongation at 61°C for 1 min for 25 cycles, and final elongation step at 61°C for 8 min. All oligonucleotide sequences are provided in S1 Table.

Pf2004/164tdTom ring stage parasites were transfected with 100 µg of the plasmid above using electroporation as previously described [51]. To select for parasites carrying the plasmids, transfected parasites were grown in presence of 2.5 µg/ml blasticidin-S deaminase for the first 10 days and then in absence of drug pressure until a stably propagating parasite population was established (approximately 4 weeks post-transfection). Clones of transgenic parasites were generated by serial dilution [52]. Successful integration in the transfected parasite population was confirmed by PCR on gDNA extracted using DNeasy blood and tissue kit (Qiagen), with primers listed in (S1 Table).

### *P. falciparum in vitro* culture and gametocyte induction

Strains used in this study were NF54 (Nijmegen, the Netherlands) and Pf2004, originally from Ghana [53]. The transgenic parasite lines NF54- and Pf2004–164TdTomato expresses the tandem red fluorescent protein TdTomato under the control of an early gametocyte promoter [10,51,54] and requires addition of 4 nM WR99210 (Jacobus Pharmaceuticals) to culture to maintain the transgene. Transgenic lines, GEXP02-GFP reporter lines were used for time course studies to track the earliest stages of gametocytogenesis [20]. The 3D7 WT, *Pf*PL-SCR1-KO and *Pf*PL-SCR1-GFP tagged lines were generously gifted by the Baum lab [28].

*Plasmodium falciparum* parasites were grown *in vitro* as previously described [55] in fresh type O+ human erythrocytes (NHS), suspended at 5% haematocrit. Culture media was composed of HEPES-buffered RPMI 1640 0.05% gentamycin, 0.0201% sodium bicarbonate (w/v) and 0.005% hypoxanthine (w/v) at pH 6.74. Serum media was supplemented with 10% human serum (Interstate Blood Bank), while serum-free medium (MFA) was prepared as previously described [56] by additional supplementation of 0.39% fatty acid-free BSA, 30 µM oleic acid, and 30 µM palmitic acid (all from Sigma-Aldrich). Cultures were kept at 37°C in a gassed chamber at 5% $CO_2$ and 1% $O_2$.

Gametocytes were produced as previously described [54]. Briefly, parasites were double synchronised using 5% sorbitol to within an 8-hour window and seeded at 1.5–2% parasitemia and 5% hematocrit. Serum media was replaced with MFA with when the cultures were between 20–30hpi. Cultures were returned to serum media after 20hrs incubation with MFA. hours, after which the cultures were returned to serum medium. Beginning 24 hours after serum medium replacement, media was changed every day with addition of heparin (230ug/ml) for 2 consecutive days to completely block reinvasion and deplete the asexual stages. Parasite development and conversion was observed by daily smearing. For time course experiments gametocytes were harvested on a daily or on alternative days for 10–11 days to capture the different morphological forms, a typical time course being day 3–4 days post invasion (4–5 days post induction) for stage I/II; day 5–6 stage II/III; day 7–8 stage III/IV and day 9–11 stage V gametocytes. On days 8–11 E64 at 10uM final concentration was added to culture media to prevent gamete activation.

### iRBC enrichment by magnetic-activated cell sorting (MACS)

We utilized the QuadroMACS or MACS midi system (Miltenyi Biotec, Bergisch Gladbach, Germany) to enrich for iRBCs (mature asexual stages and all gametocyte stages) following a modified protocol [57]. Cultures were washed once in

incomplete RPMI, resuspended to 50% hematocrit and applied to MACS LD column pre-loaded with RPMI. Gametocytes and mature asexual parasites retained in the column were eluted with incomplete RPMI and washed 3x with RPMI. Following purification, cells were counted using a hemocytometer and aliquoted for downstream assays.

## Flow cytometry

Flow cytometry was used to measure antibody binding to the surface of live iRBCs. 96-well plates were pre-coated with FACS buffer (1X PBS with 1% heat-inactivated foetal bovine serum (FBS) and 2mM EDTA) for 1 hour at room temperature or overnight at $4^0$C, to prevent non-specific binding. Aliquots of purified asexual stage trophozoites or gametocytes (resuspended in 1X PBS) were pre-incubated for 1 hr with test primary antibodies, simultaneously with glycophorin C (2ug/mL) conjugated with Alexafluor 488 (1:500). Secondary antibodies (goat anti-rabbit IgG and goat anti-mouse diluted 1:500) were added to the cells and incubated for 30 min at room temperature. Nucleus dye Vybrant DyeCycle Violet diluted 1:500 or Hoechst diluted 1:1000 (Thermo Fisher Scientific) was then added to the cells and incubated at 37°C for 30 min. The volume of each well was increased to 200uL. Plate was centrifuged at 400g for 5 min and washed 3 times with 1% FACS buffer between every staining/incubation stages. All samples were tested in triplicate.

FACS data acquisition and analysis were performed on BD FACSCelesta (BD Biosciences, NJ, USA) or MACSQuant (Miltenyi). Events were acquired to allow analysis of a minimum of 5000 infected cells per sample. Data were analysed with FlowJo software. Samples were displayed in a bivariate plot of side scatter area vs forward scatter area to gate for intact cells and forward scatter area vs height to gate for singlets. Nucleus stain was used to gate for iRBCs and TdTomato (0164/TdTomato lines), mScarlet or GFP (GEXP02 lines) was used to gate for gametocytes. Data were displayed as either frequency of IgG positive cells or geometric mean fluorescence intensity (MFI), corrected for background signal from secondary only cell staining.

## Immunofluorescence microscopy

The same antibody combinations and incubations were used for live immunofluorescence. After staining and washing, cells were mounted onto Concanavalin A (Sigma) coated hydrophobic printed well slides (EMS, 63425–05) with Vectashield (Vector Labs, Burlingame, CA, USA).

For fixed and permeabilised IFA, MACS purified mature asexual stages (36–44hpi) or gametocyte stages were washed 3x in PBS and subjected to trypsin/chymotrypsin treatment as above. Cells were fixed for 40 min in 4% paraformaldehyde/0.01% glutaraldehyde, washed 3x in PBS, permeabilized for 10 min using 0.1% Triton-X and washed 3x. GEXP07 EL antibodies (against the extracellular loop of GEXP07, kindly provided by Dr. Philippe Deterre), GEXP10 EL antibodies (against the extracellular loop of GEXP010, kindly provided by Dr. Philippe Deterre), GEXP10 antibodies (our own antibody, see [12]) or SBP-1 antibody (N-terminal rabbit, kindly provided by Dr. Tobias Spielmann) were then incubated with trypsin/chymotrypsin-treated or mock-treated iRBCs at 0.4% haematocrit, and surface binding was detected using Alexa Fluor 488 anti-IgG dye (Thermo Fisher Scientific; 1:500). Nucleus staining was done using DAOI or Hoechst. After all incubations washes were carried out with PBS supplemented with 1% FBS. Cells were mounted with Vectashield✓ (Vector Labs, Burlingame, CA, USA). Images were obtained under a 60X objective using Nikon A1R inverted confocal microscope. Images and movies were generated using Image J software 2.9.0/1.53t.

## Phagocytosis assay

Phagocytosis assays were performed as previously described [12,17]. Briefly, the assay measures phagocytic uptake of iRBCs via opsonic antibodies and Fcγ receptors on undifferentiated THP-1 cells (European collection of authenticated cell cultures), which do not express CD36 and therefore do not promote non-opsonic phagocytosis. THP-1 cells were maintained in THP-1 culture medium at $5 \times 10^5$ cells/ml in a humidified incubator at 37°C supplemented with 5% $CO_2$. iRBCs (mature asexual stages or gametocytes) were MACS purified and stained with 10 mg/ml dihydroethidium (DHE)(Sigma- Aldrich) for

30 minutes in the dark (RT). Cells were washed thrice with RPMI and adjusted to a cell density of 8.25 x 10$^6$ cells/ml. For opsonization test sera, 3.3 µL serum was added followed by 30 µL iRBC suspension in triplicates to 96-well plates previously pre-incubated with heat-inactivated new-born calf serum. For non-opsonized phagocytosis cells were kept in RPMI. Rabbit anti-RBC antibody (Abcam ab34858) was used a positive control (1:100). After 1 hour of opsonization at RT, cells were washed thrice with RPMI, re-suspended in 25 µL THP-1 medium and aliquoted into new plates (pre-incubated with NCS). 50 µL of THP-1 cells (5 x 10$^5$ cells/ml) were then added to each well and the plate was incubated for 40 min at 37 °C in a $CO_2$-supplemented humidified incubator. After phagocytosis, the plate was centrifuged at 350 x g for 5 minutes at 4°C. The supernatant was removed and 75 µL FACS lysing solution (BD Biosciences) was added for 10 minutes (RT). The plate was then spun down (4°C) and washed thrice with FACS buffer. Cells were filtered before acquisition on a MACSQuant Analyzer flow cytometer (Miltenyi Biotec) or imaged on the Nikon confocal microscope. After gating for live cells based on forward/side scatter, a gate for positive DHE fluorescence was set based on <5% cells of the negative control (no sera added). Phagocytic index (percentage of THP-1 cells that phagocytosed iRBCs) for each sample was then determined by subtracting average negative control fluorescence and dividing by average positive control fluorescence.

## Immunoelectron microscopy

For immunolabeling, the samples were fixed in phosphate buffer, pH 7.2, containing 4% freshly prepared formaldehyde. After several washes in the same buffer, they were dehydrated in ascending ethanol series and embedded in LR White resin (Agar Scientific). Ultrathin sections (70 nm thick) were obtained using an ultramicrotome (Leica Microsystems). The sections were collected on formvar-coated nickel grids and then blocked in PBS containing 3% bovine serum albumin for 1 hour. After this time, they were incubated in the presence of primary antibodies (anti-GEXP07 EL, anti-GEXP10 EL or anti-GEXP10). Then they were washed several times in blocking buffer and incubated with 10 or 15 nm gold-conjugated Protein A (Aurion). The grids were washed several times in the blocking buffer, dried and contrasted with 4% uranyl acetate, and observed using a JEOL 1200 EX transmission electron microscope operating at 80kV. Gold particles were quantified by eye as a measure of antigen labelling by the respective antibodies.

## Annexin V binding

250,000 cells or less of purified mature asexual parasites, gametocytes, unpurified rings and uninfected RBCs were used in the assay. Briefly, cells were washed 3 times in RPMI and stained Hoechst (1:1000/ 20uM) for 15 minutes at room temperature. Stained cells were washed 2 times with annexin V buffer (130-092-820 Miltenyi) and resuspended in 25ul of annexin V buffer. 2.5ul Annexin V-FITC or Annexin V-AF647 was added to each sample and incubated for 15 minutes at room temperature in the dark. The cells were washed twice with 200ul Annexin V buffer, resuspended in 100ul Annexin V buffer and run on flow cytometer or mounted on glass dishes for imaging as described above.

## Inhibition assay

**PMV treatment.** To block protein export at the onset parasites were exposed to the inhibitor WEHI-842 for 48 hours starting at the ring stages (asexual rings or gametorings) before 16hpi. To block protein export in maturing gametocytes parasite were exposed to WEHI-842 at stage II (day 5) for 48 hours. Read outs were performed at mature asexual stages, stage II and stage IV-V gametocytes.

**PS exposure inhibitors.** To test inhibitors that block PS surface flipping during asexual schizogony trophozoite stage parasites were incubator with complete media containing serial dilutions of the inhibitors for <20 hours. A subset of parasites was MACS purified at schizonts at >40hpi to measure annexin V binding by flow cytometry. The remaining subset was let to reinvade and mature for 24 hours to quantify parasite multiplication rate (PMR) post inhibitor treatment.

To block PS surface flipping at stage II gametocytes parasite were treated with the inhibitors on day 2 post induction (early stage I gametocytes) for 2 consecutive days. Readout by flow cytometry was carried out on day 5 post induction at stage II gametocytes. To block the reversal of PS surface exposure stage II-III gametocytes (day 6 post invasion) were treated with the inhibitors for 2 consecutive days. Readout by flow cytometry was carried out on day 8–9 post induction at stage IV-V gametocytes. For all the experiments above the following readouts were made, (i) % positive annexin V for PS surface detection, (ii) immune sera and the antibodies GEXP07 EL and GEXP10 EL to measure effect of inhibitors on surface antigen exposure. For one inhibitor, R5421 a further sample of the treated stage II gametocytes was allowed to mature to stage IV to measure any late effect of the inhibitor on loss of surface antigens.

## Supporting information

**S1 Fig.  Dynamics of antigen exposure A,B.** Detection of human serum reactivity in NF54 by flow cytometry (A) and live microscopy (B). C,D. GEXP10 EL (C) and GEXP07 EL (D) antibody surface reactivity in NF54. E,F. Reactivity of GEXP07 EL antibodies against live Pf2004 cells by live microscopy (E) and flow cytometry (F). G,H,I. Serum (G), GEXP10 EL (H) and GEXP07 EL (I) antibody reactivity in Pf2004 across asexual and gametocyte development by flow cytometry. Same data as in Figs 1B, 1D and S1F, respectively, but plotted as % positive cells. A-C, F-I represent the mean from three biological replicates.
(TIF)

**S2 Fig.  A major antigen pool remains internal in gametocytes.** A. IFA data. GEXP10 EL surface labelling dynamics in fixed cells, internal labelling dynamics in fixed and permeabilized cells. B. Immuno EM with representative images of asexual schizonts and stage II gametocytes. A fraction of GEXP10 protein (based on GEXP10 antibody) is present on the surface of asexual and gametocyte iRBCs. C. Flow cytometry quantification of +/-Triton permeabilised cells. Asexual (24–36hpi), GI-Iia (D4), GIIb-III (D6), GIII-IV (D8), GIV-V (D11). Panels show from top to bottom immune serum, GEXP10, GEXP10 EL and GEXP07 EL reactivity. Blue: DAPI, green: IgG, red: Tdtomato. C represents the mean from three biological replicates.
(TIF)

**S3 Fig.  Control experiments for parasite surface antigen expression and PS exposure.** A. Dynamics of internal anti-spectrin antibody labelling in live, fixed only, and fixed and Triton permeabilised cells across gametocyte development. B,C. Variation of serum (B) and annexin V (C) labelling across experiments and using different donor RBC batches (1–4) for parasite culture. A represents the mean from two (stage I) and three (stage II-V) biological replicates.
(TIF)

**S4 Fig.  Inhibition of membrane lipid remodelling affects antigen exposure and immune recognition.** A-F. Testing various inhibitors to block PS exposure or internalization. (A) Syk inhibitor, (B) PKC alpha inhibitor (PKCa) – chelerythrine chloride, (C) Flippase inhibitor - NEM, (D) Senicapoc - Gardos channel inhibitor, (E) Vanadate - ATP inhibitor, (F) Scramblase inhibitor R5421/ oxalic acid dehydrate/ ethanoininic acid. A-F represent the mean from three biological replicates.
(TIF)

**S5 Fig.  Phagocytosis assays.** A. Control experiment using asexual, immature and mature gametocyte stages. Opsonisation with immune serum (left) and with US control serum (right). B,C. Effect of PMV inhibitor on opsonisation of asexual stages and immature gametocytes using immune serum vs US control serum. Same samples as in Fig 6A and 6B respectively. A-C represent the mean from three biological replicates.
(TIF)

**S1 Table.  Oligonucleotide primers for the generation of the transgenic GEXP02-GFP line.**
(XLSX)

**S2 Table. Overview table for raw data S3–S11 Tables.**
(XLSX)

**S3 Table. Raw data Figs 1 and S1.**
(XLSX)

**S4 Table. Raw data Fig 2.**
(XLSX)

**S5 Table. Raw data Fig 3.**
(XLSX)

**S6 Table. Raw data Fig 4.**
(XLSX)

**S7 Table. Raw data Figs 5 and S4.**
(XLSX)

**S8 Table. Raw data Fig 6.**
(XLSX)

**S9 Table. Raw data S2 Fig.**
(XLSX)

**S10 Table. Raw data S3 Fig.**
(XLSX)

**S11 Table. Raw data S5 Fig.**
(XLSX)

## Acknowledgments

We would like to thank Alexander Maier (ANU, Canberra) for helpful discussions and advice, Philippe Deterre (INSERM, Paris) for CBPI (GEXP10 EL) and CBPII (GEXP07 EL) antibodies, Jake Baum (UNSW, Sydney) for *Pf*PL-SCR1-KO and *Pf*PL-SCR1-GFP parasite lines, Justin A. Boddey (The Walter and Eliza Hall Institute of Medical Research) for the PMV inhibitor, and Leandro Lemgruber (Glasgow Imaging Facility, University of Glasgow) for electron microscopy experiments.

## Author contributions

**Conceptualization:** Priscilla Ngotho, Teun Bousema, Matthias Marti.

**Data curation:** Priscilla Ngotho.

**Formal analysis:** Priscilla Ngotho.

**Funding acquisition:** Matthias Marti.

**Investigation:** Priscilla Ngotho, Kathleen Dantzler Press, Megan Peedell, William Muasya, Brian Roy Omondi, Stanley E. Otoboh, Jahiro Gomez, Teun Bousema.

**Methodology:** Priscilla Ngotho, Kathleen Dantzler Press, Megan Peedell, Karl B. Seydel, Melissa Kapulu, Miriam Laufer, Terrie Taylor, Teun Bousema.

**Project administration:** Matthias Marti.

**Resources:** Karl B. Seydel, Terrie Taylor.

**Supervision:** Priscilla Ngotho, Lorena Coronado, Melissa Kapulu, Miriam Laufer, Terrie Taylor, Teun Bousema, Matthias Marti.

**Visualization:** Priscilla Ngotho, Matthias Marti.

**Writing – original draft:** Priscilla Ngotho, Matthias Marti.

**Writing – review & editing:** Kathleen Dantzler Press, Melissa Kapulu, Miriam Laufer, Terrie Taylor, Teun Bousema.

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
