## [Decision Letter · Decision Letter 0]

30 Sep 2024

Dear Prof. Marti,

Thank you very much for submitting your manuscript "Reversible host cell surface remodelling limits immune recognition and maximizes survival of Plasmodium falciparum gametocytes" for consideration at PLOS Pathogens. As with all papers reviewed by the journal, your manuscript was reviewed by members of the editorial board. In light of the reviews obtained through Review Commons, and your submitted Revision Plan, we would like to invite the resubmission of a significantly-revised version that takes into account the reviewers' comments and this plan.

We cannot make any decision about publication until we have seen the revised manuscript and your response to the reviewers' comments. Your revised manuscript is also likely to be sent to reviewers for further evaluation.

Sincerely,

Noah S. Butler

Guest Editor

PLOS Pathogens

Dominique Soldati-Favre

Section Editor

PLOS Pathogens

Michael Malim

Editor-in-Chief

PLOS Pathogens

orcid.org/0000-0002-7699-2064

Reviewer's Responses to Questions

**Part I - Summary**

Reviewer #1: The manuscript must be rejected for formal reasons. The data described in the results section do not match the figures, and these in turn do not match the figure legends (e.g. in the results section Fig. 2D is supposed to show PS staining, the figure itself shows a diagram with different amounts of whatever and the figure legend suggests that this is an IFA image; or, Fig. S3A is supposed to show something with internal binding of spectrin antibody, but shows a diagram for annexin binding). In general, the data are not described in sufficient detail, unscientific terms are used (e.g. line 120: protein is expressed on the iRBC surface versus the protein is located on the iRBC surface). The proteins examined are referred to differently in the text than in the figures (e.g. GEXP10 versus CBPI versus CBPII; see Fig. 1F). Some figures are simply missing and called "pending" (e.g., Fig. S1D). In conclusion, it is impossible to understand the experiments and conclusions described in the manuscript and the review process is an imposition on reviewers.

**Part II – Major Issues: Key Experiments Required for Acceptance**

Reviewer #1: No further comment here

**Part III – Minor Issues: Editorial and Data Presentation Modifications**

Reviewer #1: No further comment here

PLOS authors have the option to publish the peer review history of their article (what does this mean? ). If published, this will include your full peer review and any attached files.

**Do you want your identity to be public for this peer review?** For information about this choice, including consent withdrawal, please see our Privacy Policy .

Reviewer #1: No
---

## [Decision Letter · Decision Letter 1]

9 Mar 2025

PPATHOGENS-D-24-01845R1

Reversible host cell surface remodelling limits immune recognition and maximizes survival of Plasmodium falciparum gametocytes

PLOS Pathogens

Dear Dr. Marti,

Thank you submitting your manuscript "Reversible host cell surface remodelling limits immune recognition and maximizes survival of Plasmodium falciparum gametocytes" (PPATHOGENS-D-24-01845R1) for consideration at PLOS Pathogens. As with all papers peer reviewed by the journal, your manuscript was reviewed by members of the editorial board and by several independent peer reviewers. Based on the reviews (below this email), we would like to invite the resubmission of a revised version that takes into account the reviewers' comments.

I am returning your manuscript with three reviews. All three reviewers remarked on the general scientific interest of your investigation into Plasmodium-induced remodeling of the red blood cell membrane and its influence on immune recognition and survival of the infected cells. As you can see, the reviewers each found many positives in your work, highlighting important data. All three recommend publication subject to either further experiments, analysis, or clarifications that support the main findings. After considering the manuscript and comments of the reviewers, I am therefore recommending Minor Revisions.

Of particular note in the reviewer comments:

(i) Are there approaches that complement the use of export inhibitors that might have effects on PS exposure (i.e. conditional interference of PTEX to provide another line of evidence of this central point)?

(ii) Are there opportunities to use endogenously tagged marker proteins to complement the antibody-based IFAs/EM methods to quantify localization and export? If studies that address points i and ii are not possible, please acknowledge and discuss limitations of the selected approaches.

(iii) Please expand on or clarify how the authors believe a scramblase localized to parasite internal membranes is playing a role in PS exposure in the host cell membrane.

(iv) Clarify reference to CBPI, CBP2, GEXP7 and GEXP10 within the text and Figures, as they appear to be used interchangeably throughout the manuscript.

(v) Relevant to Figure 2B, please provide representative flow cytometry plots that enable readers to understand the difference between Annexin V positive and Annexin V negative cells so that readers can understand and appreciate these important differences.

(vi) There are missing or mislabeled details in the text, figures, and figure legends. Please address these minor comments in the revised manuscript and consider opportunities to clarify the experimental design, text, and discussion, as indicated.

Although we cannot make any decision about publication until we have seen the revised manuscript and your response to the reviewers' comments, do note that your revised manuscript and responses may be sent to the reviewers for further evaluation.

Please submit your revised manuscript within 30 days May 08 2025 11:59PM. If you will need more time than this to complete your revisions, please reply to this message or contact the journal office at plospathogens@plos.org. If you would like to make changes to your financial disclosure, competing interests statement, or data availability statement, please make these updates within the submission form at the time of resubmission. Guidelines for resubmitting your figure files are available below the reviewer comments at the end of this letter.

Thank you for submitting your manuscript to PLOS Pathogens. We look forward to receiving your revised manuscript.

Kind regards,

Noah S. Butler

Guest Editor

PLOS Pathogens

Dominique Soldati-Favre

Section Editor

PLOS Pathogens

Sumita Bhaduri-McIntosh

Editor-in-Chief

PLOS Pathogens

orcid.org/0000-0003-2946-9497

Michael Malim

Editor-in-Chief

PLOS Pathogens

orcid.org/0000-0002-7699-2064

**Journal Requirements:**

1) Please ensure that the CRediT author contributions listed for every co-author are completed accurately and in full. At this stage, the following Authors/Authors require contributions: Matthias Marti. Please ensure that the full contributions of each author are acknowledged in the "Add/Edit/Remove Authors" section of our submission form.

2) Please upload a copy of Figure 5E which you refer to in your text on page 24 line 602. Or, if the figure is no longer to be included as part of the submission please remove all reference to it within the text.

3) We have noticed that you have uploaded Supporting Information files, but you have not included a complete list of legends. Please add a full list of legends for your Supporting Information files after the references list.

4) Please ensure that the funders and grant numbers match between the Financial Disclosure field and the Funding Information tab in your submission form. Note that the funders must be provided in the same order in both places as well. Currently, the order of the funders is different in both places.

5) Please provide an updated Data Availability Statement in the submission form, ensuring you include all necessary access information. If your research concerns only data provided within your submission, please write "All data are in the manuscript and/or supporting information files" as your Data Availability Statement.

**Reviewers' Comments:**

Reviewer's Responses to Questions

**Part I - Summary**

Reviewer #2: This very interesting manuscript by Ngotho et al., is a revised version of a previous submission. Both reviewers offered constructive criticism in line with my thoughts, and the authors now nicely address some of the points. Two remaining concerns that center on the manuscript’s core claims would benefit from additional thoughts and experimental validation before publication in PP (please see below)

Reviewer #3: One weakness of the study is that the statistics are poorly explained. The significance of the findings is somewhat overstated as the effect discussed is detected in less than half the infected erythrocytes. Without transmission data, the use of the findings as a starting point remains to be determined.

Reviewer #4: This is a revised manuscript previously reviewed by others. The key findings relate to the exposure of parasite surface antigens during gametocyte development and the concurrent red cell membrane lipid remodeling specifically PS. Overall, the authors have addressed the issues raised by the previous reviewers satisfactorily and the paper has therefore improved significantly.

There are a number of minor issues the authors should still consider.

**Part II – Major Issues: Key Experiments Required for Acceptance**

Reviewer #2: 1. Although there is a significant difference in the annexin binding in the scramblase KO, it is conceptually challenging to understand why a scramblase localized to parasite internal membranes is playing a role in PS exposure in the host cell membrane. Although a fascinating observation, as it stands, the mechanistic insights into this phenotype is very limited. This might be challenging to address experimentally, however, a discussion of this unusual phenotype seems warranted.

2. The correlation between PS exposure and protein export in gametocytes is solely based on inhibitors with off-target side effects on other biological processes in the parasite, which might have downstream effects on PS exposure. To circumvent this, the authors could probe into export/PS correlation by direct interference with the translocon (using, for instance, conditional inhibition of PTEX components such as EXP2 or HSP101) to provide another line of evidence of this central point.

For quantitative localization/export data, the use of endogenously tagged marker proteins instead of the provided antibody-based IFAs/EM is advised.

Reviewer #3: The authors use CBPI, CBP2, GEXP7 and GEXP10 interchangeably, which is very confusing. In figure 1F, one set of columns is labeled as CBPI, another as GEXP10. If these are different proteins, the authors should explain this clearly and define the proteins more clearly. If they are not different proteins, it would be very useful for the reader to stick with one. In some cases, the text will use one name of the protein and figure the other.

It is difficult to understand the difference between Annexin V positive and Annexin V negative (Figure 2B) without seeing a representative plot. The authors should include an example of their cytometry result so that the reader can understand how different positive and negative cells are on cytometry plot.

It is unclear what the ‘samples’ in Figure 3A represent. Are these individual experiments? Does % positive refer to the percentage of positive cells within one samples? If so, please indicate how many cells were counted per sample.

Reviewer #4: None

**Part III – Minor Issues: Editorial and Data Presentation Modifications**

Reviewer #2: (No Response)

Reviewer #3: The results in Figure 2D would be clearer if the authors false-coloured either the PS 674 or the TdTom white. Currently, the magenta and red are difficult to differentiate.

In figure 3B, the labeling of iv-vi is missing.

In the legend for Figure 3, ‘PMV’ is mentioned when ‘PMV inhibitor’ is likely meant.

Line 433: The statement that this provides a proof of concept is overstated without any transmission experiments. Only a small percentage of infected erythrocytes is affected, so it remains to be determined whether targeting PS exposure affects transmission significantly.

Line 536 1500rpm would be better described in x g.

Line 544 ‘data was’ should be ‘data were’

The NEM used for the experiment shown in Figure 5A may also affect protein transport. The authors should comment.

The difference between the control and the NEM sample in Figure 5C is very large. Is the significance of that really only *? The number of * is not defined in any figure legend, an explanation of the value of the different number of * should be added.

Note

Reviewer #4: 1. In Figure 1 the authors show that the majority of the parasite surface antigens remain internal and suggest that this may imply another function. Do the authors know whether these proteins are exported to the PVM or the host cell cytoplasm as they could then act in an immune evasion mechanism post red blood cell rupture?

2. A key experiment is the use of a PMV inhibitor that is known to impact export of exported parasite proteins. However, the data seems to indicate that the inhibitor does not prevent recognition of gametocyte by immune sera. Does this suggest that there are potentially surface antigens that are not dependent on PMV for its export - this could be explored a bit more in the manuscript.

3. Fig5 and line 266 - as far as I understand the inhibitor was added on day 3 and annexin staining was measured on day 6 and this showed no apparent reduction in staining as compared to the control. However in Figure 2 the authors show that annexin V staining is already near its peak in stage I gametocytes. This would suggest that the inhibitor might be already added to late. Can the authors clarify this?

4. In the discussion l 326 the authors suggest that inhibition of surface expression may present a potential therapeutic strategy as this would make gametocytes more accessible to immune recognition and therefore clearance. At the same time they state throughout the manuscript that surface expression of parasite antigens is important for immune recognition. While I think I understand what the authors try to indicate this is somewhat confusing.

5. There seems to be a discrepancy (line 400) between this work and the published work by Dantzler et al. in relation to the role of parasite surface antigen expression and phagocytosis - while I understand that experimental differences could account for this it would be beneficial for the reader if this is clearly described.

6. The Marti lab as well as others have previously published work that indicates differences in the rigidity of the maturing gametocyte and splenic retention. based on the work here would it make sense that the initial surface expression of parasite antigens is key for targeting the bone marrow niche and that the retention is then driven by biomechanic differences. It would be of value of the authors consider addressing these two aspects in the discussion.

PLOS authors have the option to publish the peer review history of their article (what does this mean? ). If published, this will include your full peer review and any attached files.

**Do you want your identity to be public for this peer review?** For information about this choice, including consent withdrawal, please see our Privacy Policy .

Reviewer #2: No

Reviewer #3: No

Reviewer #4: No

**Figure resubmission:**

**Reproducibility:**

To enhance the reproducibility of your results, we recommend that authors of applicable studies deposit laboratory protocols in protocols.io, where a protocol can be assigned its own identifier (DOI) such that it can be cited independently in the future. Additionally, PLOS ONE offers an option to publish peer-reviewed clinical study protocols. Read more information on sharing protocols at https://plos.org/protocols?utm_medium=editorial-email&utm_source=authorletters&utm_campaign=protocol

---

## [Editor Report · Decision Letter 2]

8 Apr 2025

Dear Prof. Marti,

We are pleased to inform you that your manuscript 'Reversible host cell surface remodelling limits immune recognition and maximizes survival of Plasmodium falciparum gametocytes' has been provisionally accepted for publication in PLOS Pathogens.

Best regards,

Noah S. Butler

Guest Editor

PLOS Pathogens

Dominique Soldati-Favre

Section Editor

PLOS Pathogens

Sumita Bhaduri-McIntosh

Editor-in-Chief

PLOS Pathogens

orcid.org/0000-0003-2946-9497

Michael Malim

Editor-in-Chief

PLOS Pathogens

orcid.org/0000-0002-7699-2064

The authors have addressed the Reviewers' comments and questions and appropriately revised the manuscript.
---

## [Editor Report · Acceptance letter]

Dear Prof. Marti,

We are delighted to inform you that your manuscript, "Reversible host cell surface remodelling limits immune recognition and maximizes survival of Plasmodium falciparum gametocytes," has been formally accepted for publication in PLOS Pathogens.

Best regards,

Sumita Bhaduri-McIntosh

Editor-in-Chief

PLOS Pathogens

orcid.org/0000-0003-2946-9497

Michael Malim

Editor-in-Chief

PLOS Pathogens

orcid.org/0000-0002-7699-2064